# Improvement in Visceral Adipose Tissue and LDL Cholesterol by High PUFA Intake: 1-Year Results of the NutriAct Trial

**DOI:** 10.3390/nu16071057

**Published:** 2024-04-04

**Authors:** Nina Marie Tosca Meyer, Anne Pohrt, Charlotte Wernicke, Laura Pletsch-Borba, Konstantina Apostolopoulou, Linus Haberbosch, Jürgen Machann, Andreas F. H. Pfeiffer, Joachim Spranger, Knut Mai

**Affiliations:** 1Department of Endocrinology and Metabolism, Charité—Universitätsmedizin Berlin, Corporate Member of Freie Universität Berlin, Humboldt-Universität zu Berlin, and Berlin Institute of Health, Charitéplatz 1, 10117 Berlin, Germany; nina.meyer@charite.de (N.M.T.M.);; 2NutriAct-Competence Cluster Nutrition Research Berlin-Potsdam, Arthur-Scheunert-Allee 114-116, 14558 Nuthetal, Germany; 3Institute of Biometry and Clinical Epidemiology, Charité—Universitätsmedizin Berlin, Corporate Member of Freie Universität Berlin, Humboldt-Universität zu Berlin, and Berlin Institute of Health, Charitéplatz 1, 10117 Berlin, Germany; 4BIH Charité Junior Clinician Scientist Program, BIH Biomedical Innovation Academy, Berlin Institute of Health, Charité—Universitätsmedizin Berlin, Charitéplatz 1, 10117 Berlin, Germany; 5BIH Charité Junior Digital Clinician Scientist Program, BIH Biomedical Innovation Academy, Berlin Institute of Health, Charité—Universitätsmedizin Berlin, Charitéplatz 1, 10117 Berlin, Germany; 6Institute for Diabetes Research and Metabolic Diseases (IDM), Helmholtz Center Munich, University of Tübingen, Otfried-Müller-Str. 10, 72076 Tübingen, Germany; juergen.machann@med.uni-tuebingen.de; 7Section on Experimental Radiology, Department of Diagnostic and Interventional Radiology, University Hospital Tübingen, Otfried-Müller-Straße 12/1, 72076 Tübingen, Germany; 8German Center for Diabetes Research, Ingolstaedter Landstrasse 1, 85764 Neuherberg, Germany; 9Department of Human Nutrition, German Institute of Human Nutrition, Potsdam-Rehbruecke, Arthur-Scheunert-Allee 114-116, 14558 Nuthetal, Germany; 10DZHK (German Centre for Cardiovascular Research), Partner Site Berlin, Potsdamer Str. 58, 10785 Berlin, Germany; 11Max Rubner Center for Cardiovascular Metabolic Renal Research, Hessische Str. 3-4, 10115 Berlin, Germany

**Keywords:** visceral adipose tissue, polyunsaturated fatty acid, NutriAct, healthy aging, LDL cholesterol

## Abstract

We assessed the effect of a dietary pattern rich in unsaturated fatty acids (UFA), protein and fibers, without emphasizing energy restriction, on visceral adipose tissue (VAT) and cardiometabolic risk profile. Within the 36-months randomized controlled NutriAct trial, we randomly assigned 502 participants (50–80 years) to an intervention or control group (IG, CG). The dietary pattern of the IG includes high intake of mono-/polyunsaturated fatty acids (MUFA/PUFA 15–20% E/10–15% E), predominantly plant protein (15–25% E) and fiber (≥30 g/day). The CG followed usual care with intake of 30% E fat, 55% E carbohydrates and 15% E protein. Here, we analyzed VAT in a subgroup of 300 participants via MRI at baseline and after 12 months, and performed further metabolic phenotyping. A small but comparable BMI reduction was seen in both groups (mean difference IG vs. CG: −0.216 kg/m^2^ [−0.477; 0.045], partial η^2^ = 0.009, *p* = 0.105). VAT significantly decreased in the IG but remained unchanged in the CG (mean difference IG vs. CG: −0.162 L [−0.314; −0.011], partial η^2^ = 0.015, *p* = 0.036). Change in VAT was mediated by an increase in PUFA intake (ß = −0.03, *p* = 0.005) and induced a decline in LDL cholesterol (ß = 0.11, *p* = 0.038). The NutriAct dietary pattern, particularly due to high PUFA content, effectively reduces VAT and cardiometabolic risk markers, independent of body weight loss.

## 1. Introduction

Increased body weight, hyperglycemia and hyperlipidemia represent major risk factors for mortality from non-communicable diseases, particularly in regions of high socioeconomic status and in participants of higher age [1]. Overweight and obesity are causal factors driving metabolic disturbances including dyslipidemia and insulin resistance (IR) [2]. Especially abdominal obesity affects cardiometabolic risk profile [3], even in individuals of normal weight [4]. Abdominal fat consists of subcutaneous (SCAT), intra-abdominal or visceral (VAT) and ectopic (hepatic, pericardial, skeletal, intramuscular) fat depots [4,5] of which VAT increases cardiovascular disease (CVD) risk to the greatest extent [5,6], by promoting hepatic and peripheral IR, atherogenic lipid profile and inflammation [4,5,7]. An effective tool to reduce VAT is weight reduction, which is accompanied by cardiometabolic risk reduction [5]. Body weight regain is a common phenomenon after lifestyle-induced weight loss interventions [8], diminishing the beneficial cardiometabolic effects [7]. More importantly, in the elderly, a cohort particularly at risk for CVD [1], the effect of weight loss on metabolic risk reduction is uncertain and dramatic weight loss may even have adverse effects with respect to loss of muscle mass [9]. Therefore, guidelines on weight loss intervention in older adults recommend approaches with only moderate caloric restriction and minimization of loss in fat-free mass [9].

It is well known that nutrition can also modify CVD risk and VAT accumulation beyond weight loss [10]: while a diet high in simple carbohydrates and low-fiber diet might increase VAT [10] and cardiometabolic risk factors [11], a low-carb diet (35% carbohydrates of daily energy intake) was seen to be more beneficial regarding cardiovascular risk profile in the context of a weight loss strategy with similar weight loss between groups [12]. Diets high in animal protein have been shown to reduce VAT [13], accompanied even by a decline in IR [13]. Plant-based proteins specifically lower LDL-C levels [14]. The intake of poly- and mono-unsaturated fatty acids (PUFA, MUFA) from marine sources or out of olive oil was found to protect from diet-induced VAT accumulation, IR and dyslipidemia [10], although the beneficial effects of PUFA was not reported by others [15].

The Mediterranean diet (MeD), rich in UFA, plant-based and sea foods, low in animal products and with high intake of extra virgin olive oil, became well-known for its beneficial effect on cardiovascular risk [16]. The effect of MeD on VAT, however, is rather unclear [17]. Generally, most of the data on VAT-reducing effects through nutrition derive from studies that include only small groups, are short-termed and do not apply MRI as the gold standard for measurement of VAT [17].

Therefore, we aimed to address the question if a dietary pattern based on high amount of MUFAs and PUFAs, plant protein, fibers and complex carbohydrates can specifically reduce VAT and thereby cardiometabolic risk in the elderly. This project represents a prespecified 12-month interim analysis of the NutriAct trial, a 3-year multicenter randomized controlled intervention trial, comparing the mentioned dietary pattern with a control diet. According to our knowledge, this is the first study addressing this research question in the elderly within a randomized long-term trial.

## 2. Materials/Subjects and Methods

### 2.1. Study Design and Participants, Inclusion and Exclusion Criteria

The NutriAct trial is a randomized controlled lifestyle-intervention trial, which compares two different dietary patterns concerning their long-term effects on health over an intervention period of 36 months. It is a multicenter trial, conducted from 2016 to 2021 in Berlin (Metabolic Research Unit of the Clinic of Endocrinology and Metabolism, Charité—Universitätsmedizin Berlin) and Potsdam (Human Study Center of the German Institute of Human Nutrition Potsdam-Rehbruecke). The RCT has been described in detail elsewhere [18]. In short, a total of 502 participants (319 women, 183 men), aged between 50 and 80 years, were included. A further inclusion criterion was the presence of at least one pre-specified feature of unhealthy aging, including arterial hypertension, heart failure, existent CVD, evidence of cognitive dysfunction or impaired muscular strength. Exclusion criteria were the presence of an acute severe CVD, uncontrolled hypertension, type 1 diabetes or insulin-treated type 2 diabetes, untreated active endocrine disorders, malignant diseases, life expectancy < 1 year, severe liver or kidney disease, gait disorders, systemic infection, immune diseases, oral glucocorticoid therapy, severe food allergy, eating disorder, severe malabsorptive disease, psychiatric disorders or severe abuse of drugs and/or alcohol. Family members or individuals residing in the same household were assigned to the same group. Further specific details on the named inclusion and exclusion criteria and the sample size calculation can be found in the Appendix A. Every participant gave written informed consent before being included in the study. Electronic case report forms (eCRF, REDCap^®^14.0.17) were used to record data.

The study protocol was authorized by the Institutional Review Board of the Charité Medical School (EA1/315/15). Good clinical practice according to the Declaration of Helsinki was applied throughout the trial. Registration of the study was undertaken in advance at the German Clinical Trials Register (DRKS00010049).

Here, we analyzed data of a subcohort with MRI data available both at baseline and after 1 year (*n* = 300) in an ITT analysis. The trial’s profile is depicted in Appendix A.

### 2.2. Intervention

After baseline characterization, participants were randomly allocated to either the intervention group (IG) or control group (CG, 1:1). Details on randomization can be found in the Appendix A and were also described previously [18].

Participants in the IG were asked to follow the specific NutriAct food pattern focusing on higher intake of UFA and plant protein, with daily intake of 35–40% of total energy (% E) fat with ≤10% E saturated fatty acids (SFA), 15–20% E MUFA and 10–15% E PUFA; 15–25% E proteins (primarily plant protein); and 35–45% E carbohydrates with ≥30 g fiber. The participants in the IG were provided with specially manufactured foods rich in the aforementioned components (cf. Appendix A; [19]). They received 11 group sessions of 4–8 participants over the 12 months, including dietary counseling, cooking and lifestyle courses. Constant physical activity was recommended.

The CG was treated with usual care following recommendations by the German nutrition society (DGE) [20], based on daily intake of 30% E fat (MUFA ≥ 10% E, PUFA 7–10% E, SFA ≤ 10% E), 15% E protein, 55% E carbohydrates and ≥30 g fiber. These participants received three sessions of nutritional counseling during 12 months and were also provided with some conventional foods free of charge (see Appendix A for details).

Nutritional counseling was performed by professional dieticians. Data on nutrient intake were collected 10–14 days before each phenotyping visit, applying open food records on three consecutive days, one weekend day included. The nutrient calculation software Prodi^®^ 6.5 Expert (Nutri-Science GmbH, Freiburg, Germany) was used to analyze and convert the nutritional data. During the intervention, weight loss was not intended in both groups, and participants were asked to aim for body weight maintenance. A detailed description of the interventional protocol has been provided previously [18].

### 2.3. Anthropometric and Metabolic Assessment

We performed phenotyping of participants at baseline as well 1 year after the start of dietary intervention. Phenotyping started at 8:00 a.m. at each visit and included, among others, anthropometric measurements and routine laboratory assessments from fasting blood draws. We analyzed HOMA-IR for estimation of insulin resistance, according to Matthews et al. [21]. Data collection is described more in detail elsewhere [18].

### 2.4. Quantification of Adipose Tissue Depots and Intrahepatic Lipid Content

At baseline and after 12 months, adipose tissue compartments were analyzed using axial MRI technique while intrahepatic lipids (IHL) were quantified via proton MR spectroscopy (^1^H-MRS), using a 1.5-T whole-body scanner (Magnetom Avanto^®^, Siemens Healthcare, Erlangen, Germany). According to existing protocols published by Machann et al. [22], an axial T1-weighted fast spin-echo technique was applied to measure abdominal fat compartments. An automated segmentation algorithm based on fuzzy clustering was used for quantification of VAT volume from femoral head to thoracic diaphragm, indicated in liters [23]. Non-visceral adipose tissue (NVAT) was quantified analogous to VAT between femoral head and shoulders and represents the remaining trunk fat depots apart from VAT. For quantification of IHL, a single-voxel stimulated echo acquisition mode (STEAM) technique was applied according to existing protocols [22]. Details to IHL quantification were already reported [18] and can be found in the Appendix A. Collected imaging/spectroscopic data were analyzed in a blinded manner by an experienced medical physicist (JM) at the University Hospital of Tübingen.

### 2.5. Biochemical Analyses

After blood draw and subsequent centrifugation of samples, they were immediately analyzed or stored at −80 °C. Standard laboratory methods using an ABX Pentra 400 (HORIBA ABX SAS, Montpellier, France) were applied for assessment of lipid parameters, triacylglycerol (TG), total cholesterol (TC) and HDL cholesterol (HDL-C). The Friedewald formula (FW) was used for calculation of LDL cholesterol (LDL-C). With a TG level above 3.5 mmol/L, this was not applicable in one subject for which LDL-C is thus missing. A Cobas Mira^®^ Analyzer (Roche Diagnostics, Mannheim, Germany) was used for measurement of glucose in fluoride plasma (S-Monovette^®^ GlucoEXACT; Sarstedt, Nuembrecht, Germany). Serum insulin was analyzed by commercial enzyme-linked immunosorbent assay (Mercodia, Uppsala, Sweden; intra-assay CV 2.8–4.0%, inter-assay CV 4–5%).

### 2.6. Outcomes and Statistical Analyses

This project focused on the interventional effects on adipose tissue compartments and concomitant metabolic changes within the first 12 months of the trial. The main outcome of this prespecified ITT analysis is change in VAT, while change in NVAT and metabolic parameters are secondary outcomes.

Data are presented as mean with standard deviation (SD), median and limits of interquartile range (IQR: 25th–75th percentile), depending on distribution, or as group sizes and proportions. To test within-group differences, we used Student’s *t*-test for normally distributed parameters (one-tailed test) and Wilcoxon Signed-Rank Test for non-normally distributed parameters. To test between-group differences, we used ANCOVA models, adjusting for respective baseline values, applying Bonferroni adjustment for multiple comparisons. In the case of VAT and NVAT, we respectively created a basic model (adjusting for baseline values only, models 1; 1.1) and adjusted models, additionally adjusting for age and sex (models 2; 2.1) as well as change in IHL (models 3; 3.1) or change in lipid-lowering medication (models 4; 4.1). For all ANCOVA models, the mean difference (MD) was indicated as mean difference between CG and IG. We checked homogeneity of regression slopes by examining interaction terms between covariates and the categorical variable. It was given in all except four ANCOVA analyses (indicated). We checked and excluded outliers using leverage values (Huber) and Cook distances. In two cases, one outlier was excluded in each case (leverage value 0.21 or Cook distances > 1; indicated). We tested for normality of residuals via normality tests. In case it was not given, we applied bootstrapping with 1000 samples. This did not significantly change results in any case. Homogeneity of variances was tested by Levene’s tests (based on median). In the case of violation, we indicated it but assumed robustness of the ANCOVA models given approximately equal group sizes. We adjusted for change in antidiabetic or lipid-lowering medication when applicable.

As changes in adipose tissue compartments were given as absolute changes, we applied the same to IHL despite skewness of data.

For correlation analyses, we applied Pearson correlations or partial correlations in case of adjustment. We confirmed linearity between all variables via scatterplots. Given the large sample size of *n* = 300, we assumed normal distribution.

We performed mediation analyses using the approach of Baron and Kenny [24].

If the two-sided *p*-value was <0.05, results were considered statistically significant. 

As the sample size for the NutriAct trial was calculated for the study’s primary outcome (cf. Appendix A), these results must be interpreted as exploratory rather than confirmatory. 

Statistical analyses were performed using SPSS^®^ 26 and higher (SPSS Inc., Chicago, IL, USA), and the mediation models were computed in SPSS^®^ AMOS 25^®^.

### 2.7. Characteristics of the Study Group

Baseline characteristics including nutritional habits of the participants in the IG and CG were not different and are depicted in Table 1.

## 3. Results

### 3.1. Baseline Associations between VAT, Dietary Intake and Cardiometabolic Ris Markers

At baseline, low protein, carbohydrate and fiber intake (by trend) were related to higher VAT (Appendix A). Higher VAT was associated with higher BMI, waist circumference, estimates of insulin resistance and triglyceride levels as well as lower HDL-C (Appendix A). The inverse correlations with TC and LDL-C disappeared after adjustment for intake of lipid-lowering medication. All other associations were stable even after adjustment for antidiabetic or lipid-lowering medication, where applicable.

### 3.2. Nutritional Changes within the First Year of Intervention

As already reported about the entire NutriAct cohort [19], also in this subsample, a substantial increase in PUFA, MUFA and protein as well as a decline in SFA were seen in both IG and CG, while less strong in the CG. Intake of fibers increased, and intake of carbohydrates decreased in the IG and stayed stable in the CG (Table 2). There was a significant between-group difference for change in intake of all macronutrients between the IG and CG, with a particularly strong effect for SFA and PUFA (Table 2).

### 3.3. Effects of the Dietary Intervention on Adipose Tissue Compartments

Overall, weight loss within 12 months was small and similar between both groups (Table 3). VAT was significantly decreased after 12 months in the IG (mean change −0.23 ± 0.74 L, *p* < 0.001, Cohen’s d −0.31), while no change was observed in the CG (mean change −0.07 ± 0.64 L, *p* = 0.166). Change in VAT differed significantly between the groups after adjusting for baseline VAT values (basic model 1: MD = −0.162 L, 95%-CI: [−0.314; −0.011], *F*(1, 297) = 4.459, *p* = 0.036, partial η^2^ = 0.015; Figure 1a) and also after additional adjusting for age and sex (model 2: MD = −0.158 L; 95%-CI: [−0.310; −0.007], *F*(1, 295) = 4.245, *p* = 0.040, partial η^2^ = 0.014). The difference in change in VAT between groups appeared to be independent of the change in liver fat content as between-group difference persisted after adjusting for the change in IHL (model 3: MD = −0.162 L; 95%-CI: [−0.306; −0.018], *p* = 0.028, partial η^2^ = 0.018; homogeneity of regression slopes not given for the interaction term intervention group × change in IHL).

In both groups, a stronger decline in VAT was seen in those who had a higher increase in PUFA (ρ = −0.177, *p* = 0.003) and protein intake (ρ = −0.116, *p* = 0.049). No relationship to change in MUFA, SFA, fiber and carbohydrate intake could be revealed (Appendix A).

The reduction in NVAT was more pronounced in the IG (−0.93 ± 1.34 L, *p* < 0.001) compared to CG (−0.52 ± 1.26 L, *p* < 0.001), with a significant between-group difference in change in NVAT (model 1.1: MD = −0.407 L; 95%-CI: [−0.685; −0.128], *F*(1, 292) = 8.256, *p* = 0.004, partial η^2^ = 0.027); Figure 1b). This significant difference persisted also after adjusting for age and sex (model 2.1: MD = −0.412 L; 95%-CI: [−0.691; −0.132], *F*(1, 290) = 8.407, *p* = 0.004, partial η^2^ = 0.028) and also after adjusting for change in IHL between baseline and 1 year (model 3.1: MD = −0.405 L; 95%-CI: [−0.687; −0.122], *F*(1, 266) = 7.947, *p* = 0.005, partial η^2^ = 0.029; homogeneity of regression slopes not given). These effects persist after adjustment for lipid-lowering medication (change in VAT: MD = −0.157 L; 95%-CI: [−0.309; −0.005], *p* = 0.043, partial η^2^ = 0.014; change in NVAT: MD = −0.424 L; 95%-CI: [−0.704; −0.145], *p* = 0.003, partial η^2^ = 0.030).

A stronger increase in PUFA intake was also associated with a higher decline in NVAT (ρ = −0.227, *p* < 0.001), while no relationship to changes in protein, MUFA, SFA, fiber or carbohydrate intake was found (Appendix A).

### 3.4. Effects of the Dietary Intervention on Cardiometabolic Risk Factors

A comparable reduction in IHL and HOMA-IR was seen in both groups, whereas significantly stronger reductions in TC and LDL-C and by trend in HDL-C were observed in the IG (Table 3).

Improvements in HOMA-IR and lipid profile were significantly associated with the decline in VAT (Table 4a).

The decline in NVAT was associated with a decrease in HOMA-IR, TG and an increase in HDL-C. In contrast, NVAT changes were not linked to changes in TC and LDL-C (Table 4b).

### 3.5. Relationship between Changes in Nutrient Intake, Adipose Tissue Compartments and Changes in LDL-C

We analyzed which factors drove the between-group difference in the adipose tissue compartments and the major cardiometabolic risk factor LDL-C. Mediation analyses revealed that increased PUFA intake in the IG was a substantial mediator of VAT reduction (ß = −0.03), while the decline in VAT had an independent effect on LDL-C improvement (ß = 0.11; Figure 2a; Appendix A).

Reduction in NVAT was also mediated by the increase in PUFA intake (ß = −0.06), but no relationship to LDL-C levels was detected (Figure 2b; Appendix A).

Increase in protein intake did not mediate the interventional effect on VAT (adjusted *p* = 0.072) despite a significant association with the reduction in VAT and a significant between-group difference in protein intake. There was also no association between change in protein intake and change in LDL-C (adjusted *p* = 0.752).

## 4. Discussion

### 4.1. Effects on Adipose Tissue Compartments

Our data demonstrate a long-term effect of a dietary pattern based on a high intake of UFA, protein and fiber, specifically on VAT reduction, which was not seen under the control diet despite a comparable mild weight loss in both groups. Interestingly, this effect on VAT seems to be independent of the improvement in liver fat, which might indicate different underlying mechanisms. Comparable findings were reported in a weight loss intervention trial on women with obesity and high vs. low IHL content at baseline: a hypocaloric diet induced a weight loss of 8%, with s decrease in intra-abdominal, subcutaneous as well as intrahepatic fat depots. However, the reduction in IHL did not correlate with the decline in intra-abdominal fat or SCAT, indicating different regulations for these fat stores [25].

In our study, the reduction in VAT within the IG was primarily mediated by increased PUFA intake, while modulation of MUFA, protein and fiber intake was not of major impact. In contrast, earlier studies on PUFA-rich diets found effects on SCAT [26] or overall trunk fat [27] but not on VAT or IHL [28]. Thus, specifically the effect of PUFA intake on VAT reduction in humans seems to be a novel finding, as it is not suggested by the aforementioned studies. These were, however, of shorter duration, smaller sample sizes or based on a more specific cohort composition. The achieved relative increase in PUFA intake was smaller compared to our study (23% [26] vs. 69%), or only *n*-3 PUFA were used [26,27], while we mainly applied rapeseed oil consisting of both *n*-6 and *n*-3 PUFA.

On a molecular basis, PUFA might reduce adipose tissue mass by various mechanisms [29]: they affect energy homeostasis, amplifying energy expenditure, partly through promotion of thermogenesis in adipose tissue but also through effects on the neuroendocrine system. PUFAs also inhibit lipid synthesis, primarily by suppressing relevant enzymes, such as fatty acid synthase and stearoyl-CoA desaturase, and lipogenesis on the one hand and enhance lipid oxidation and lipolysis on the other hand. Furthermore, PUFAs affect adipocyte differentiation, growth and, thus, the process of fat storage. *n*-3 PUFA are shown to reduce adipocyte hypertrophy specifically in intra-abdominal fat depots [30].

In our study, a change in protein intake was also significantly associated with a decline in VAT, but it did not seem to be a major driver of this improvement. This is in line with a meta-analysis of 54 RCTs comparing high- and low-protein diets that reported a beneficial but only small effect of high protein intake on fat mass in >4300 analyzed participants [14]. In the aforementioned study by Markova et al. [13], in which an effect of protein intake on VAT reduction was reported, two isocaloric high-protein diets (30 EN%), animal vs. plant protein, were compared for 6 weeks in participants with T2DM. In contrast to our study, the daily protein intake was relatively high, and the significant VAT reduction occurred in the animal but not in the plant protein group, which might explain the different results.

In contrast to VAT, NVAT reduced in both groups, although it was also more pronounced in the IG. Analogous to VAT, this was mediated by increased PUFA intake.

The decrease in NVAT was relatively larger compared to that of VAT, and this was observed in both groups. Nevertheless, in analogy to VAT, it was more pronounced in the intervention group. This might suggest that the response of NVAT exhibits slightly greater sensitivity to dietary interventions.

### 4.2. Effects on Lipid and Glucose Metabolism

In our study, we observed quite unexpected changes in cholesterol levels, traditionally considered less responsive to dietary interventions compared to markers such as BMI, HOMA and triglycerides, which in our study did not differ significantly between groups.

The increased PUFA intake leads to a reduction in NVAT and VAT. However, only changes in visceral adipose tissue distribution directly affect the change in LDL-C levels. Beneficial effects of a PUFA-enriched diet on LDL-C in humans have been reported since the late 1980s, independently of total fat intake [31]—explainable by the modulatory effects of PUFA on enzyme activities and gene expression of proteins involved in lipid metabolism [32]. Marine *n*3-PUFA are reported to reduce the synthesis of apolipoprotein (B) and thus lower LDL-C levels and to stimulate membrane phospholipid composition [33]. The Mediterranean diet in the PREDIMED study, focusing on high MUFA content, did not affect LDL-C levels when compared to a low-fat diet [34]. However, a recent meta-analysis of clinical trials demonstrated that replacement of carbohydrates with PUFA and MUFA both leads to a decrease in LDL-C, while the effect is stronger with PUFA [35]. Our data suggest that a decrease in VAT might be an important mediator of PUFA-induced LDL-C reduction.

Surprisingly, the stronger VAT loss in the IG was not accompanied by a more intensive decrease in HOMA-IR, although both changes were associated. This is in line with data from another lifestyle intervention study in adolescents with obesity [36], in which VAT was associated with and even predictive of IR, but no difference in ∆HOMA was observed between groups differing in magnitude of VAT loss. This supports the hypothesis that metabolic dysfunction through VAT and IHL are based on different mechanisms [37]. Nevertheless, we could not entirely exclude that the observed decline in IHL and VAT might be too small to affect HOMA-IR, as other studies indicated that quite intense reductions in body weight, IHL and total or visceral fat mass are required to achieve an effect on HOMA [38].

Moreover, other macronutrients might diminish the positive impact of VAT reduction on IS. The KANWU study, assessing the effect of a SFA substitution by MUFA on IS in 162 healthy persons for 3 months, showed that a positive metabolic effect of MUFA replacement was diminished when total fat intake exceeded 37% E [39]. In the present study, we observed a mean total daily fat intake of >37% at baseline and >39% at month 12 within the IG, which might explain the missing improvement in IS. Also an increased ratio of *n*-6 to *n*-3 PUFA > 4:1 might reduce the beneficial health effects of *n*3-FA [33] and promote IR [40]. In the PREDIMED trial, however, a better glycemic control was achieved under olive oil (rich in MUFAs, PUFA: *n*6:*n*3 > 4:1) compared to nut supplementation [41]. As we primarily used rapeseed oil (also rich in MUFA, PUFA: *n*6:*n*3 < 4:1), this aspect does not seem to play a major role. Also, a high protein intake (25–50 E%, mixed plant and animal protein) was shown to even induce IR, possibly due to higher protein expression of ribosomal subunit serine kinase 6–1 (S6K1), leading to reduced glucose uptake [42]. Although we promoted the intake of fibers, which could compensate this effect [42], a negative impact of high protein intake on IS cannot be ruled out in our study. 

Finally, as in our study, a whole dietary pattern was applied for the long-term, rather than short-term, replacement of single macronutrients, providing a more realistic setting but possibly weaker effects.

### 4.3. Strengths and Limitations

Whether *n*3- or *n*6-PUFAs are more beneficial or harmful to human health is a topic of constant debate. An important limitation of this study is that no differentiation of *n*3- and *n*6-PUFAs was undertaken. We could only assume that intake of both PUFAs were increased, as we primarily used rapeseed oil. Ethnic differences in body fat distribution and its metabolic implications are known [43], an aspect which was not covered by our study. We did not assess the modulatory impact of changes in muscle mass in relation to adipose tissue. Also, the possible impact of physical activity was not considered, but we aimed for equalization between groups by design. Dietary incompliance cannot be entirely excluded. However, we assessed dietary intake repeatedly by food records, a well-accepted assessment of macronutrient intake in trials [44].

The study has multiple important strengths. First is the prospective nature of the study. As an RCT, it provides robust results, supported by the large cohort size and long intervention period. The assessment of adipose tissue compartments was conducted via MRI technique, which represents the most exact method to date [45]. The application of a dietary *pattern* represents a realistic approach implementable in real life.

## 5. Conclusions

In conclusion, the NutriAct diet, containing high amounts of PUFA and protein, effectively reduces VAT without promoting body weight loss, leading to a reduction in LDL-C. It therefore represents a reasonable option for the prevention of cardiometabolic diseases in elderly persons.

## Figures and Tables

**Figure 1 nutrients-16-01057-f001:**
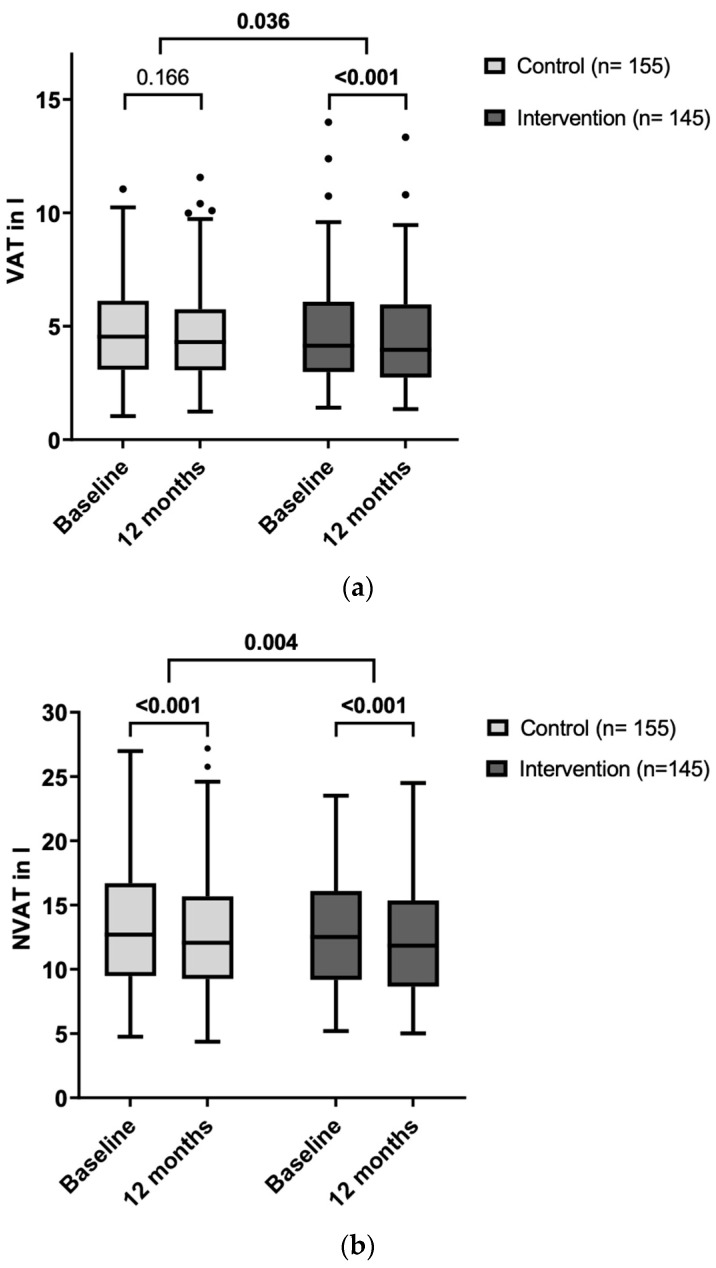
Abdominal fat volume at baseline and after 12 months, CG vs. IG. (**a**) VAT. (**b**) NVAT. Boxplots: (**a**) VAT and (**b**) NVAT at baseline and after 12 months within control and intervention group, respectively. VAT, visceral adipose tissue. NVAT, non-visceral adipose tissue (abdomen). *p*-values for between-group comparisons were adjusted for baseline values (basic model). VAT, visceral adipose tissue; NVAT, non-visceral adipose tissue; (●) outliers.

**Figure 2 nutrients-16-01057-f002:**
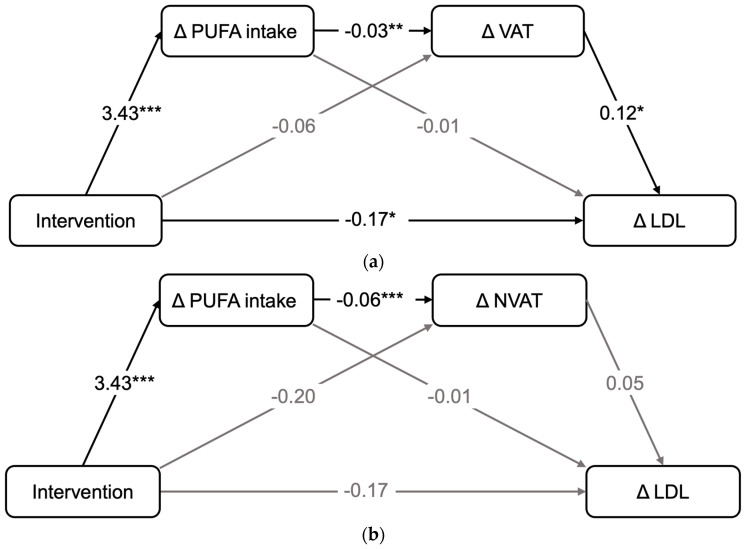
Statistical diagrams of the mediation analyses. (**a**) Intervention—∆PUFA—∆VAT—∆LDL. (**b**) Intervention—∆PUFA—∆NVAT—∆LDL. Mediation models of direct and indirect effects of the NutriAct intervention on changes in LDL-C, mediated by ∆PUFA-intake and/or ∆VAT (**a**)/∆NVAT (**b**). Adjusted for age, sex, baseline VAT, baseline LDL-C values and lipid-lowering medication. *** significant at α = 0.001 ** significant at α = 0.01; * significant at α = 0.05. Black connection line: effect significant. Grey connection line: effect non-significant. VAT, visceral adipose tissue; PUFA, polyunsaturated fatty acids; LDL-C, low-density lipoprotein cholesterol; NVAT, non-visceral adipose tissue.

**Table 1 nutrients-16-01057-t001:** Baseline participants’ characteristics and nutritional data (*n* = 300).

Characteristics	Control Group		Intervention Group	
	Value	N	Value	N
Demographics				
Females (%)	63.2	98	62.8	91
Age (years)	65 ± 7	155	66 ± 7	145
<60 y. (%)	20	31	16.6	24
≥60 and <65 y. (%)	20	31	17.9	26
≥65 and <70 y. (%)	34.2	53	36.6	53
≥70 and <75 y. (%)	18.1	28	18.1	28
≥75 y. (%)	7.7	12	11.0	16
Patients with diabetes mellitus type 2 (%)	11.0	17	9.7	14
Patients with arterial hypertension (%)	82.6	128	82.1	119
Patients with hepatic steatosis (%)	38.1	59 ^a^	35.9	52 ^b^
Patients with metabolic syndrome (%)	40.6	63 ^c^	44.1	64 ^d^
Anthropometrics				
BMI (kg/m^2^)	29.2 ± 4.7	155	29.1 ± 4.5	145
Waist circumference (cm)	98.7 ± 11.5	155	98.5 ± 11.2	145
TAT (L)	18.2 ± 6.2	155	18.1 ± 6.3	145
VAT (L)	4.7 ± 2.2	155	4.6 ± 2.3	145
NVAT (L)	13.3 ± 5.1	155	13.4 ± 5.1	145
IHL (%)	4.1 [1.6; 9.4]	155	3.9 [1.5; 8.0]	145
Glycemic metabolism				
HOMA-IR	1.8 [1.2; 2.9]	155	1.6 [1.1; 2.7]	145
Lipid profile				
TC (mmol/L)	5.4 ± 1.1	155	5.4 ± 1.1	145
LDL-C (mmol/L)	3.32 ± 1.0	155	3.4 ± 0.9	144
HDL-C (mmol/L)	1.4 ± 0.3	155	1.4 ± 0.3	145
TG (mmol/L)	1.2 [1.0; 1.6]	155	1.2 [1.0; 1.6]	145
Nutritional data				
Energy intake (kcal·d^−1^)	1967 ± 524	151	1988 ± 539	142
Protein intake (g·kg^−1^·d^−1^)	1.0 ± 0.3	151	1.0 ± 0.3	142
Carbohydrates intake (% E·d^−1^)	40.0 ± 6.8	151	40.8 ± 7.2	142
Fiber intake (g·d^−1^)	22.3 ± 7.9	151	23.2 ± 7.9	142
Saturated fatty acids intake (% E·d^−1^)	15.9 ± 3.3	151	15.7 ± 3.9	142
MUFA intake (% E·d^−1^)	13.2 ± 2.7	151	13.0 ± 2.8	142

Data are described as mean ± standard deviation for normally distributed variables, median [interquartile range] for non-normally distributed continuous variables and as *n* (%) for categorical variables. Metabolic syndrome is defined according to ATP criteria. ^a^ out of 151 patients with valid data. ^b^ out of 142 with valid data. ^c^ out of 153 patients with valid data; 1.3% (*n* = 2) missing due to missing information about permanent medication to baseline. ^d^ out of 143 patients with valid data; 1.4% (*n* = 2) missing due to missing information about permanent medication to baseline. Abbreviations: y, years; BMI, body mass index; TAT, total adipose tissue; VAT, visceral adipose tissue; NVAT, non-visceral adipose tissue; IHL, intrahepatic lipid content; HOMA-IR, homeostatic model assessment-Insulin resistance; TC, total cholesterol; LDL-C, low-density lipoprotein cholesterol; HDL-C, high density lipoprotein cholesterol; TG, triacylglycerols. Dietary components are described as % of the total daily energy intake, except for protein and fiber intake, which are described as g·kg^−1^ bodyweight and g·d^−1^ respectively. Abbreviations: MUFA, monounsaturated fatty acids.

**Table 2 nutrients-16-01057-t002:** Differences in macronutrient changes (baseline 1 year) between intervention and control group, adjusted to baseline values.

	Intervention Group			Control Group			Intervention vs. Control Group		
Parameters	Mean ± SE	*p*-Value ^a^	N	Mean ± SE	*p*-Value ^a^	N	Mean Difference [95% CI ^a^]	*p*-Value ^b^	Partial η^2^
∆ Energy intake (kcal·d^−1^)	−122.4 ± 34.4	**0.040**	141	−226.2 ± 33.5	**<0.001**	149	103.8 [9.4; 198.3]	**0.031**	0.016
∆ Protein intake (g·kg^−1^·d^−1^)	0.11 ± 0.02	**<0.001**	141	−0.04 ± 0.02	**0.017**	149	0.16 [0.10; 0.22] ^c^	**<0.001**	0.079
∆ Protein intake (% E·d^−1^)	2.35 ± 0.29	**<0.001**	141	0.88 ± 0.28	**0.014**	149	1.47 [0.67; 2.27] ^c^	**<0.001**	0.044
∆ Carbohydrate intake (% E·d^−1^)	−3.58 ± 0.50	**<0.001**	141	−0.00 ± 0.48	0.362	149	−3.58 [−4.95; −2.20] ^d^	**<0.001**	0.084
∆ Fiber intake (g·d^−1^)	3.63 ± 0.64	**<0.001**	141	−0.91 ± 0.63	0.158	149	4.55 [2.78; 6.32] ^c,d^	**<0.001**	0.082
∆ Fatty acid intake (% E·d^−1^)	1.76 ± 0.52	**0.004**	141	−0.94 ± 0.51	**0.048**	149	2.70 [1.26; 4.13]	**<0.001**	0.045
∆ MUFA intake (% E·d^−1^)	0.56 ± 0.23	**0.013**	141	−0.67 ± 0.22	**0.002**	149	1.22 [.59; 1.86]	**<0.001**	0.048
∆ PUFA intake (% E·d^−1^)	4.2 ± 0.28	**<0.001**	141	0.72 ± 0.27	**0.003**	149	3.52 [2.75; 4.28] ^c,e^	**<0.001**	0.221
∆ SFA intake (% E·d^−1^)	−3.28 ± 0.29	**<0.001**	141	−0.95 ± 0.28	**0.001**	149	−2.33 [−3.12; −1.54]	**<0.001**	0.105

Dietary components are given as % of total daily caloric intake, except for fibers, which are described as g/d. Univariate ANCOVA. Test of the effect of intervention group, based on the linearly independent pairwise comparisons among the estimated marginal means. Adjustment for respective baseline values and covariates appearing in the model are evaluated at the following baseline values: energy intake (kcal·d^−1^) = 1978.6; protein intake (g·kg^−1^·d^−1^) = 1.0 g; protein (%) = 16.4; carbohydrate (%) = 40.4; fibers (g/day) = 22.7; fatty acids (%) = 37.7; MUFAs (%) = 13.1; PUFAs (%) = 6.0; saturated fatty acids (%) = 15.8. ^a^ *p*-value for within-group comparison using Student’s *t*-test (one-tailed) for parametric and Wilcoxon Signed-Rank Test for non-parametric values. ^b^ *p*-value for between-group difference, adjusted for multiple comparisons: Bonferroni. ^c^ homogeneity of variances is not given. ^d^ homogeneity of regression slopes (group × carbohydrate intake to baseline/group × fiber intake to baseline) not given. ^e^ 1 extreme outlier (leverage value > 0.2) was excluded. Abbreviations: kcal, kilocalorie; MUFA, monounsaturated fatty acids; PUFA, polyunsaturated fatty acids; SFA, saturated fatty acids. Significant *p*-values are in bold type.

**Table 3 nutrients-16-01057-t003:** Effects on cardiometabolic risk factors.

	Intervention Group			Control Group			Intervention vs. Control Group		
Parameters	Mean ± SE	*p*-Value ^a^	N	Mean ± SE	*p*-Value ^a^	N	Mean Difference [95% CI ^a^]	*p*-Value ^b^	Partial η^2^
∆ BMI (kg/m^2^)	−0.71 ± 0.10	<0.001	145	−0.50 ± 0.09	<0.001	155	−0.22 [−0.48; 0.05]	0.105	0.009
∆ waist circumference (cm)	−2.13 ± 0.63	<0.001	145	−1.73 ± 0.60	0.003	155	−0.40 [−2.11; 1.31]	0.643	0.001
∆ IHL (%)	−1.94 ± 0.25	<0.001	132	−1.58 ± 0.24	<0.001	145	−0.36 [−1.05; 0.33]	0.302	0.004
∆ HOMA-IR	−0.56 ± 0.08	<0.001	138	−0.50 ± 0.08	<0.001	149	−0.06 [−0.27; 0.16] ^c^	0.597	0.001
∆ TC (mmol/L)	−0.39 ± 0.07	<0.001	144	−0.11 ± 0.07	0.114	155	−0.29 [−0.47; −0.10]	**0.002**	0.031
∆ LDL-C (mmol/L)	−0.28 ± 0.06	<0.001	143	−0.04 ± 0.06	0.311	155	−0.24 [−0.40; −0.08] ^d^	**0.004**	0.028
∆ HDL-C (mmol/L)	−0.06 ± 0.01	<0.001	144	−0.02 ± 0.01	0.058	155	−0.04 [−0.07; 0.00] ^d^	0.058	0.012
∆ TG (mmol/L)	−0.11 ± 0.04	0.004	144	−0.09 ± 0.04	0.001	155	−0.02 [−0.11; 0.08]	0.753	<0.001

Univariate ANCOVA. Adjusted means and standard error given for each group. Test of the between-group differences comparing IG vs. CG, based on the linearly independent pairwise comparisons among the estimated marginal means. Adjustment for respective baseline values. Covariates appearing in the model are evaluated at the following baseline values and changes in medication (baseline to 1 year), respectively: BMI (kg/m^2^) = 29.1; HOMA-IR = 2.2 and change in antidiabetic medication = 1.0; TC (mmol/L) = 5.4 and change in lipid-lowering medication = 0.1; LDL-C (mmol/L) = 3.4 and change in lipid-lowering medication = 0.1; HDL-C (mmol/L) = 1.4 and change in lipid-lowering medication = 0.1; TG (mmol/L) = 1.3 and change in lipid-lowering medication = 0.1. ^a^ *p*-value for within-group comparison using Student’s *t*-test (one-tailed) for parametric and Wilcoxon Signed-Rank Test for non-parametric values. ^b^ *p*-value for between-group difference, adjusted for multiple comparisons: Bonferroni. ^c^ 1 outlier excluded (Cook distance > 1). ^d^ homogeneity of variances is not given. Abbreviations: BMI, body mass index; IHL, intrahepatic lipids; IG, intervention group; CG, control group; HOMA-IR, homeostatic model assessment-insulin resistance; TC, total cholesterol; LDL-C, low-density lipoprotein cholesterol; HDL-C, high density lipoprotein cholesterol; TG, triacylglycerols. Significant *p*-values are in bold type.

**Table 4 nutrients-16-01057-t004:** (a) Correlations between change in VAT and changes in cardiometabolic risk markers between baseline and 12 months within the entire subcohort (*n* = 300). (b) Correlations between change in NVAT and changes in cardiometabolic risk markers between baseline and 12 months within the entire subcohort (*n* = 300).

Metabolic Risk Factors	Pearson Correlation Coefficient	*p*-Value for Correlation with Delta VAT	N
(a)			
∆ HOMA-IR	0.245	**<0.001**	297
∆ TC (mmol/L)	0.162	**0.005**	299
∆ LDL-C (mmol/L)	0.151	**0.009**	298
∆ HDL-C (mmol/L)	−0.106	0.068	299
∆ TG (mmol/L)	0.207	**<0.001**	299
(b)			
∆ HOMA-IR	0.186	**0.001**	292
∆ TC (mmol/L)	0.076	0.192	294
∆ LDL-C (mmol/L)	0.088	0.135	293
∆ HDL-C (mmol/L)	−0.156	**0.007**	294
∆ TG (mmol/L)	0.122	**0.036**	294

Pearson correlation analysis of entire study population. HOMA-IR, homeostatic model assessment-insulin resistance; TC, total cholesterol; LDL-C, low-density lipoprotein cholesterol; HDL-C, high density lipoprotein cholesterol; TG, triacylglycerols. Significant *p*-values are in bold type.

## Data Availability

The datasets generated during and/or analyzed during this study and the study protocol are available on request from the corresponding author due to (privacy). The study design and study protocol are already publicly accessible.

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
