# Peer review of "Improvement in Visceral Adipose Tissue and LDL Cholesterol by High PUFA Intake: 1-Year Results of the NutriAct Trial"

_nutrients, 2024, doi:10.3390/nu16071057_

Round 1

Reviewer 1 Report

Comments and Suggestions for Authors

Manuscript Number: nutrients-2882798

Title: Improvement of visceral adipose tissue and LDL cholesterol by high PUFA intake: 1-year results of the NutriAct trial

Reviewer decision: Minor modifications

Reviewer comments:

The authors assessed the effect of a dietary pattern rich in unsaturated fatty acids (UFA), protein and fibers on visceral adipose tissue (VAT) and cardiometabolic risk profile.

Participants included in this study were randomly allocated to either the intervention group (IG) or control group (CG, 1:1). Participants in the IG followed the specific NutriAct food pattern: higher intake of UFA and plant protein, with daily intake of 35-40% of total energy  (%E) fat with ≤ 10 %E saturated fatty acids (SFA), 15-20 %E MUFA and 10-15 %E PUFA; 15-25 %E proteins (primarily plant protein); 35-45%E carbohydrates with ≥30 g fiber. The CG was treated with usual care following recommendations by the German nutrition society, based on daily intake of 30 %E fat (MUFA ≥ 10 %E, PUFA 7-10 136 %E, SFA ≤ 10 %E), 15 %E protein and 55 %E carbohydrates and ≥30 g fiber.

Overall it is a well-written manuscript, with a good experimental design, and the results showed that VAT significantly decreased in the IG but remained unchanged in the CG, and change in VAT was mediated by increase in PUFA intake and induced a decline in LDL cholesterol. Finally, the authors concluded that the NutriAct dietary pattern, particularly due to high PUFA content, effectively reduces VAT and cardiometabolic risk markers, independent of body weight loss. Ultimately, the authors came to the conclusion that the NutriAct dietary pattern, especially because of its high PUFA content, efficiently lowers VAT and cardiometabolic risk indicators, independent of body weight loss.

Minor comments:

  1. In the randomized controlled NutriAct trial, 502 people were randomly placed in two groups (page1, rows 37–38), according to the authors' abstract. However, in the current investigation, data from only 300 participants were included in the analysis. The authors are asked to reconsider to rewrite in the Abstract section the number of participants take into account in this study.
  2. In the Results section, Paragraph 3.2, the authors mentioned that substantial increase of PUFA, MUFA and protein as well as a decline in SFA was seen in both IG and CG (page 6, row 238). How those changes in the dietary behavior have been evaluated? Did the participants indicate that they followed the NutriAct pattern or did they fill out a form describing the quantity of dietary components they consumed from those listed in this study?
  3. The legend of Figure 1 states that the adipose tissue mass was depicted as a box plot; however, in the Materials and Methods section ( Paragraph 2.4. – page 4, row 162) the MRI scans show that the VAT volume (in liter) was measured and subsequently examined using statistical tests.
  4. Could the NVAT adipose tissue depot, which was evaluated by MRI in this investigation, be regarded as subcutaneous adipose tissue?
  5. Can the authors furthermore supply the IHL measures following a 12-month follow-up? IHL is a biomarker of insulin resistance, and the VAT and IHL are highly correlated with cardiometabolic risk factors.

Reviewer 2 Report

Comments and Suggestions for Authors

I’ve read with attention the paper of Meyer et al. that is of sure interest. The background and aim of the study have been clearly defined. The methodology applied is correct, the results are deeply described and reliable, and adequately discussed. I have no ethical concerns regarding experiments, nor on plagiarism or publication ethics. The lenght of the paper is adequate to its scientific content and the cited references correct in number and quality. I only would suggest the authors to add a short comment on the unexpected change of TC and LDL (usually less responsive to diet than BMI, HOMA and TG, that here don't changed compared to the control group).

Reviewer 3 Report

Comments and Suggestions for Authors

The Article ‘Improvement of Visceral Adipose Tissue and LDL Cholesterol by High PUFA Intake: 1-Year Results of the NutriAct Trial’ raises very important issues. It shows how even small changes in the profile of nutrients consumed (mainly PUFA) can affect the visceral adipose tissue (VAT) and cardiometabolic risk profile.

The abstract is written correctly, as well as the introduction. The experiment was correctly planned and executed. It can be seen that the experiment described is only a small part of a larger project between several research centres. Table S2 should be in the manuscript.

Unfortunately, I have a lot of objections to the Results chapter:

Subsection 3.1: characteristics of the study group in my opinion should be in the Methodology chapter.

Lines 239-240; 268-270 – Is it allowed to write about results that have not been shown?

Please consider whether tables S3 and S4 should not be in the main manuscript.

The description should be under the figure number and title, not above. The description of figures and tables should be distinguished from the main text.

TABLE 3 – in which group?

Figure 1 - What do the little dots and squares mean?

The Results chapter should be reviewed once again by all authors.

The discussion is correctly written.

The article is very interesting, and the results should be published, but only after corrections. Please also verify the article from an editorial point of view.

Round 2

Reviewer 3 Report

Comments and Suggestions for Authors

The article's topic is interesting, the experience was correctly planned and executed. After corrections, the results are clearer and valuable information has been added. However, I still think part of it is that the section on patient selection should be in the methods section. In addition, the authors did not pay attention to editorial issues (for example, the title of the figure is above it, and different font sizes in the text). With such a number of authors, I question whether they all saw and approved the final version.
